# Gesture Recognition Achieved by Utilizing LoRa Signals and Deep Learning

**DOI:** 10.3390/s25051446

**Published:** 2025-02-27

**Authors:** Peihao Zhang, Baofeng Zhao

**Affiliations:** Taiyuan University of Technology, Taiyuan 030024, China; 2023521308@link.tyut.edu.cn

**Keywords:** gesture recognition, LoRa, deep learning, feature extraction, signal processing

## Abstract

This study proposes a novel gesture recognition system based on LoRa technology, integrating advanced signal preprocessing, adaptive segmentation algorithms, and an improved SS-ResNet50 deep learning model. Through the combination of residual learning and dynamic convolution techniques, the SS-ResNet50 model significantly enhances the extraction capability of multi-scale gesture features, thereby augmenting the classification accuracy. To counter environmental noise and static interferences, an adaptive segmentation approach based on sliding window variance analysis is introduced in the research. This method effectively increases data diversity while preserving the specific components of gestures. Experimental outcomes indicate that the system exhibits strong robustness in cross-scenario and cross-device tests, with an average recognition accuracy of over 95% for six gestures. The system’s low power consumption, long-distance communication, and strong anti-interference capabilities offer broad prospects for its application in complex environments, particularly in resource-constrained scenarios such as underground mine gesture monitoring and remote control in dynamic environments and other practical applications. This study demonstrates the feasibility of gesture recognition systems based on LoRa technology and provides a new solution for low-power, long-distance non-contact gesture recognition.

## 1. Introduction

Gesture recognition has garnered considerable attention in recent years due to its extensive application in human–computer interaction, intelligent environments, and remote control systems. Nevertheless, traditional gesture recognition approaches based on wearable sensors or computer vision, despite their outstanding performance under controlled conditions, encounter numerous constraints such as high power consumption, strong reliance on the environment, and complex hardware. These deficiencies are particularly conspicuous in dynamic, noisy, or resource-limited scenarios, highlighting the urgent need for non-contact, low-power, and robust alternatives.

Gesture recognition based on wireless signals presents a promising solution to these issues. For example, technologies like WiFi [1] and RFID can detect gestures by exploiting signal variations, yet they often perform inadequately in complex environments: WiFi systems are significantly influenced by dynamic interferences, resulting in a substantial deterioration of classification performance in noisy circumstances; while RFID [2] systems, although featuring low power consumption, are restricted by short-range communication capabilities and are susceptible to multipath interferences.

In contrast, LoRa [3] technology (Long Range) possesses distinctive advantages in terms of low power consumption, long distance, and anti-interference capability, making it highly suitable for gesture recognition tasks in complex scenarios. Nonetheless, the exploration of LoRa in dynamic environments and real-time recognition performance in existing studies remains limited.

In comparison with the signal processing methods in current research, the novelty of this study in the domain of LoRa gesture recognition lies in the further introduction of activity segmentation technology based on the traditional LoRa signal processing framework, in conjunction with an improved SS-ResNet50 model, which significantly enhances the dynamic adaptability and classification accuracy of the system. Distinct from traditional signal preprocessing and classification methods, our activity segmentation approach effectively enhances the extraction of dynamic gesture features, while the improved model, combined with dynamic convolution technology, further elevates the flexibility and robustness of feature extraction and classification. Experimental results indicate that the system exhibits a high accuracy rate of over 95% in cross-scenario and cross-device tests, demonstrating its outstanding performance in complex environments.

The main contributions of this study are summarized as follows:The improved SS-ResNet50 model was put forward, combined with dynamic convolution technology, which significantly improved the performance of LoRa signal feature extraction and classification. We developed an adaptive segmentation approach that utilizes variance-based analysis to enhance data diversity while retaining gesture-specific characteristics, effectively addressing environmental noise and interference.An adaptive segmentation method was developed, which enhances data diversity through variance-based analysis while effectively preserving the features of dynamic gestures and reducing the impact of noise interference on recognition results. We verified the feasibility of using LoRa signals for long-distance and low-power gesture recognition, highlighting its potential for practical application in resource-constrained environments.The robustness and universality of the system were validated through indoor experiments, particularly exhibiting excellent adaptability in dynamic and complex environments, and the recognition accuracy remained at a relatively high level. This system is characterized by low power consumption and high robustness, offering technical support and research directions for future applications in resource-constrained circumstances, such as long-distance gesture recognition or underground emergency communication.

## 2. Relevant Work

### 2.1. Contact-Based Gesture Recognition

Research on gesture recognition dates back to the 1980s and has evolved significantly over the decades. Sensor-based gesture recognition primarily includes gesture recognition using data gloves and inertial sensors. Sun Fuchun et al. utilized an inertial data glove equipped with 18 IMUs to capture hand movements [4]. Gao Weihua developed a graphene-based data glove with 10 sensors to achieve precise gesture motion perception [5]. Joseph DelPreto coupled a resistive sensor network and accelerometers in a wearable device, deploying a Long Short-Term Memory (LSTM) neural network that achieved 92.8% accuracy in real-time streaming predictions [6]. Zhang Xiaoliang combined a wearable armband and a customizable pressure sensor array in a smart glove, achieving an overall classification accuracy of 77.78% across 10 consecutive gestures [7]. Zhang Peng designed a flexible pressure sensor with stable mechanical performance and high sensitivity, maintaining excellent mechanical properties under a wide range of pressure loads [8]. While these sensor-based methods provide high sensitivity, their high costs and cumbersome wearable designs limit user adoption and application scenarios.

### 2.2. Non-Contact Gesture Recognition

Non-contact gesture recognition approaches can be broadly categorized into computer vision-based and wireless signal-based methods.

L. Brethes proposed a color segmentation method using a watershed algorithm to track hand contours for gesture recognition [9]. Zhang Renyan refined optical flow estimation, focusing on feature extractors and intermediate feature enhancements to improve recognition performance [10]. Gao Qing combined RGB, depth, and 3D skeletal data using a weighted fusion method based on an improved 2D hand pose estimation model, enhancing gesture recognition reliability [11]. Hao Liang introduced a multi-scale feature learning network with Expand-Squeeze-Aggregate (ESA) modules and task-specific prediction branches, achieving notable classification performance [12]. Li Yanko developed a dual-stream framework incorporating temporal and spatial streams, employing a Multi-Scale Fusion Module (MFM) to integrate features, significantly boosting accuracy [13].

Wireless signal-based methods use technologies such as infrared, lasers, radio frequencies, and WiFi for gesture recognition. Xue Ding proposed a WiFi-based system with endpoint detection and subcarrier algorithms, achieving a 93.67% accuracy rate for six gestures [14]. Jiang Xiao’s TransTracky employed Otsu for background noise separation and cloud transfer learning for instance adaptation across users, improving recognition under varying conditions [15]. Qin Yuxi developed a direction-agnostic gesture recognition method using WiFi devices, achieving a 92.38% accuracy rate for five gestures by mitigating directional variations with a circular antenna setup [16]. Xiong Fei leveraged RFID for gesture recognition, treating gesture-induced interference as fingerprint features and achieving a 95.5% accuracy using a tag array to normalize phase signals and RSSI [17].

Recent advancements in gesture recognition have explored various methods to improve accuracy and robustness. Alyami and Luqman (2024) introduced the Swin-MSTP framework, which utilizes the Swin Transformer for spatial feature extraction and a multi-scale temporal perception (MSTP) module for time-wise feature extraction, achieving superior performance in continuous sign language recognition (CSLR) tasks compared to existing models [18]. Amangeldy et al. (2023) proposed a dynamic gesture variability recognition method based on Convolutional Neural Networks (CNNs), which automatically collects spatiotemporal features of gestures, significantly improving classification accuracy to 96% by considering both static and dynamic gesture properties [19]. Finally, Deng et al. (2024) developed the Skeleton-based Multi-feature Learning (SML) method, combining multi-feature aggregation and a Self Knowledge Distillation Guided Adaptive Residual Decoupled Graph Convolutional Network (SGAR-DGCN) for efficient feature extraction, achieving state-of-the-art performance on the WLASL and AUTSL datasets [20]. These studies highlight the increasing effectiveness of integrating advanced deep learning techniques, such as transformers and graph convolutional networks, with skeleton and spatiotemporal data for high-accuracy gesture recognition.

### 2.3. LoRa-Based Applications

LoRa technology, with its strengths in long-range communication, low power consumption, and strong anti-interference capabilities, has found extensive application in human activity recognition, healthcare monitoring, and environmental sensing. These applications leverage LoRa’s unique characteristics to overcome challenges in complex and resource-constrained environments. Below, we summarize key studies that showcase the versatility of LoRa technology in various domains, particularly in human activity recognition, laying the groundwork for its potential in gesture recognition. Specific applications are shown in Table 1.

In summary, LoRa technology has demonstrated significant potential in human activity and gait recognition due to its excellent signal transmission performance and environmental adaptability. However, studies on LoRa-based gesture recognition remain limited, particularly when leveraging LoRa’s unique characteristics for high-accuracy gesture recognition in complex scenarios. This study addresses these challenges by proposing an innovative LoRa-based gesture recognition method tailored for such applications.

## 3. System Design

This paper introduces LoSDS, a gesture recognition system consisting of a LoRa data preprocessing module and a gesture recognition module. We will provide a detailed description of it in the following section. The technical choices of this system were carefully designed to meet specific application requirements. Data preprocessing employs Short-Time Fourier Transform (STFT) to generate spectrograms, combined with Differential Signal Processing (DSP) to effectively extract activity-related features, ensuring signal reliability. The introduction of dynamic convolution significantly enhances the model’s ability to extract complex gesture features, improving recognition accuracy. Furthermore, the use of LoRa technology, with its long-range communication and low power consumption, enables efficient and stable communication in complex scenarios. These technical choices not only enhance the overall performance of the system but also establish a strong foundation for its application in diverse environments.

### 3.1. LoRa Data Preprocessing

#### LoRa Data Analysis

LoRa is a physical layer technology that uses Chirp Spread Spectrum (CSS) modulation, which is a linear frequency modulation (LFM) signal waveform represented by a complex exponential.(1)$$\;Tx(t)=e^{j2\pi f_{c}t+j\pi kt^{2}}=\cos(2\pi f_{c}t+\pi kt^{2})+j\cdot \;\sin(2\pi f_{c}t+\pi kt^{2})$$

Among them, $f_{c}$ is the center frequency, k=B/T is the sweep frequency of the linear chirp signal, B is the bandwidth of a linear chirp signal, and T is the period time of a linear chirp signal. According to the work of ZHANG et al. [3], it is assumed that a chirped signal is generated at the node:(2)$$s\left(t\right)=e^{j\pi kt^{2}}$$

If the signal reaches the receiving end via N different paths, the signal at the receiving end can be represented as:(3)$$Rx(t)=e^{j\pi kt^{2}+j2\pi ftt}\sum_{n=1}^{N}a_{n}(t)e^{-j2\pi f_{\epsilon }\tau_{n}(t)}$$

Herein, $a_{n}(t)$ denotes the attenuation factor of the nth path $\theta_{c}=2\pi \Delta ft$, $\tau_{n}(t)$ indicates the propagation delay of the signal on the nth path, $\Delta f=f_{c}-f_{c}^{\prime }$ is the CFO (Carrier Frequency Offset) resulting from the desynchronization of the clocks of the transmitter and the receiver, $\theta_{s}$ is the phase error caused by the SFO (Sampling Frequency Offset). The phase errors generated by $\theta_{c}$ and $\theta_{s}$ have a considerable influence on the actual value of the phase.

The causes of Carrier Frequency Offset (CFO) and Sampling Frequency Offset (SFO) lie in the desynchronization of the clocks between the transmitter and the receiver, resulting in the signal’s phase readings becoming unreliable. To obtain consistent CFO and SFO values, the clock oscillators of the transmitter and receiver must be synchronized. To address this issue, we equipped the receiving end with two antennas, effectively mitigating the random phase offset. Additionally, we observed that while CFO and SFO significantly affect the phase of LoRa signals during propagation, their impact on frequency is relatively minimal. This insight allows us to classify and recognize gestures by leveraging the distinct frequency variations induced by gesture movements as distinguishing features.

### 3.2. LoRa Data Conversion

To analyze the frequency changes induced by different gestures, it is necessary to convert the original LoRa data into a time-frequency plot that captures these variations over time. In our system, the input data is a complex array containing I/Q dual channels, which must first be transformed into the real domain to generate a continuous time-domain signal sequence. Subsequently, the Short-Time Fourier Transform (STFT) is applied to map the time-domain signal onto the time-frequency plane, enabling simultaneous observation of signal variations in both time and frequency. This approach provides an accurate depiction of the signal’s frequency changes over time, forming the basis for our subsequent analysis and classification.(4)$$STFT_{x}(t,\omega )=\int_{-\infty }^{\infty }x(\tau )w^{*}(\tau -t)e^{-j\omega \tau }d\tau$$

Among them, ω represents the angular frequency, w(t) represents the window function, and w(t) represents the conjugate. The meaning of this formula is: at the moment t, place the window function w(t) at position t, truncate the signal, and then perform a Fourier transform on the truncated signal.

### 3.3. Employ Signal Differences to Retain Gesture Components

In order to further improve the recognition accuracy, we use the method of signal difference to remove the static component and retain the gesture component, shown in Figure 1 below:

In a localization environment, if no other motion changes occur, the signal received by the receiver (RX) is a static vector resulting from the reflection of the signal transmitted by the transmitter (TX). This static vector remains relatively stable and can be expressed as follows:(5)Qx1(t)=mv1+mv2+mv3+mv4

If gestures occur, they inevitably generate varying reflective waveforms, causing some static vectors to transition into dynamic vectors. At this point, the receiver (RX) captures a combination of the unchanged static vector component and the dynamic vector induced by the gesture. The signal received by RX can be expressed as follows:(6)Qx2(t)=mv1+mv2+nv3+mv4

From the comparison of the formulas, it is evident that a constant vector sum remains unchanged before and after a gesture change. Our focus is on the varying component, as the frequency changes caused by gestures are relatively minor compared to the constant static vector. Moreover, based on the properties of the Short-Time Fourier Transform (STFT), addition or subtraction of signal values in the time domain corresponds to addition or subtraction of instantaneous frequencies in the frequency domain.(7)Q′x=nv3−mv3

Consequently, subtracting the sum of static vectors effectively mitigates the environmental influences, thereby enhancing the recognition system’s accuracy and adaptability. The signal processing results are shown in Figure 2.

### 3.4. Data Augmentation Through Adaptive Segmentation

After retaining the gesture component by using the signal difference, the recognition ability of the system has been enhanced. However, there is still some noise remaining in the signal waveform that has not been eliminated. The frequency changes caused by gesture transformation are typically low, while the noise frequency contained in the signal is much higher. Thus, within a signal, there will be multiple sets of signal samples representing the same gesture change. If these samples are superimposed, the phase change can be enlarged, and the noise can be eliminated. Hence, we propose an adaptive segmentation algorithm for sub-sampling. For a consecutive signal image, a portion is selected at intervals as a sample to conduct sub-sampling of the entire change process. For the numerical values of N and Δt, the condition that the sub-sampled samples must contain the same motion information should be fulfilled. Set the size of the sliding window to i and the step size to $V_{i}$, and perform the sliding window operation on the processed data. For each window position $X_{d}$, calculate the data variance within the window i, where V represents the data within the window starting from position j. The calculated variance values are stored in sequence $M_{j}$.(8)$$V_{i}=\mathrm{Var}(X_{d}[i:i+w])$$

$V_{j+k}$ represents the variance at position j+k, and these means are stored in sequence M, after which the mean sequence is sorted to obtain the maximum value, threshold $T=M_{\max}\times p$, which should be chosen as N = 7 and Δt = 0.01 s in our experiment, N represents the number of sliding window segments, *p* represents the dimensionality of features extracted from each segment, M is the total number of segments, and T is the overall dimensionality of the feature matrix. To balance the trade-off between feature coverage and data redundancy, the value of N was optimized through preliminary experiments. Results indicated that N = 7 ensures sufficient feature extraction while minimizing redundant data. Hence, N = 7 was selected as the sliding window segmentation parameter in this study.

Finally, we extract, align, and stack the sub-sampled groups that have been segmented to achieve the effect of data augmentation. The effect of data augmentation is shown in Figure 3. The effect of activity segmentation is shown in Figure 4.(9)$$M_{j}=\frac{1}{N}\sum_{k=0}^{N-1}V_{j+k}$$

## 4. Gesture Recognition

### 4.1. ResNet-50

ResNet [27] has been widely utilized in various feature extraction tasks. In theory, increasing the depth of a deep learning network enhances its representational capacity. However, as the depth of a CNN network exceeds a certain threshold, further deepening the network layers does not improve classification performance. Instead, it results in slower convergence and even a decline in accuracy. While enlarging the dataset can mitigate overfitting to some extent, it does not necessarily improve classification accuracy or performance. In this study, we select ResNet-50 as our backbone network. ResNet-50 addresses the vanishing gradient problem in deep network training through residual learning. Comprising 50 layers, it includes fundamental components such as convolutional layers, pooling layers, batch normalization, and activation functions. Its core innovation lies in the introduction of residual blocks, which utilize skip connections to directly transmit input to the output. This mechanism allows the network to learn the residuals between inputs and outputs, significantly facilitating the training of deeper networks and improving both performance and representational capacity.

### 4.2. Dynamic Convolution Module

The dynamic convolution module [28] is a technology that enhances traditional convolution operations by adaptively adjusting the weights of the convolution kernel in response to the distinct features of the input. The dynamic convolution module dynamically generates the convolution kernel during each forward propagation, enabling the model to possess greater flexibility when handling different samples. This module typically encompasses a weight generation network that generates the weights of the convolution kernel based on the global information of the input features, thereby allowing the same convolution layer to accommodate a wide variety of input data. The dynamic convolution module is shown in Figure 5.

The process of obtaining a set of weight values can be represented as:(10)$$\pi_{k}=softmax(FC(ReLu(FC(GAP(x)))))$$

Specifically, $0\leq \pi_{k}\leq 1,\sum_{k=1}^{K}\pi_{k}\left(x\right)=1$. The weight values and attention weight values are added together to form the following aggregated convolutional kernel:(11)$$\tilde{W}=\sum_{k=1}^{K}\pi_{k}(x)\tilde{W_{k}}$$

Using the same attention weight to aggregate biases:(12)$$\tilde{b}=\sum_{k=1}^{K}\pi_{k}\left(x\right)\tilde{b_{k}}$$

### 4.3. SS-Resnet50 Model

We have developed the SS-ResNet50 model by extending ResNet-50 with dynamic convolution. This model combines the adaptability of dynamic convolution with the deep architecture of ResNet-50, enabling it to better handle diverse input variations. The model retains the stability and trainability of the residual structure while making feature extraction more flexible, thereby improving the overall system performance.

During training, the initial convolution layer of SS-ResNet50 performs preliminary feature extraction on the input images. The extracted intermediate features are then passed to the dynamic convolution module, where a global feature vector is used to generate dynamic convolution kernels. These kernels are subsequently employed in adaptive convolution operations, which are combined with residual connections for efficient feature extraction. The model stacks multiple layers of dynamic convolution and residual blocks to extract high-level features. Finally, global average pooling and fully connected layers are used for classification predictions. Additionally, the dynamic convolution kernel generation process and the convolution parameters of SS-ResNet50 are continuously optimized during training through backpropagation.

The technical choices of this system were carefully designed to meet specific application requirements. The data preprocessing module employs Short-Time Fourier Transform (STFT) and Differential Signal Processing (DSP) to ensure the reliability and accuracy of feature extraction, particularly in handling noise and interference in complex environments. The introduction of dynamic convolution enhances the model’s sensitivity to gesture features, enabling higher recognition accuracy in complex gesture scenarios. LoRa technology was selected for its long-range communication and low power consumption, making it especially suitable for gesture recognition in resource-constrained and dynamic environments. These choices not only optimize system performance but also provide robust support for its application in diverse scenarios.

## 5. Experiment

### 5.1. Experimental Setup Experimental Scenarios and Data Collection

The hardware setup of the gesture recognition system consists of a pair of LoRa transmitters and receivers. The transmitting end comprises an STM32 development board (ALIENTEK, Guangzhou, China) and a Semtech SX1276 module (ALIENTEK, Guangzhou, China), equipped with a directional antenna to improve transmission distance and signal stability. The receiving end includes a USRP X310 (Ettus, Guangzhou, China) with two directional antennas, and LoRa signals are collected using LabVIEW software (2024 Q2). The LoRa signal operates at a carrier center frequency of 915 MHz, with a chirp bandwidth of 125 kHz and a sampling rate of 1 MHz at the receiver. Training experiments were conducted using PyTorch (v.3.7) on a high-performance PC featuring an Intel Core i9-12900K CPU (Santa Clara, CA, USA), NVIDIA GeForce RTX 4070 GPU, and 64 GB of RAM (Santa Clara, CA, USA).

### 5.2. Experimental Scenarios and Data Collection

Experimental scenario: The experiment was carried out in an indoor laboratory of 5 m by 10 m. As depicted in the figure, the transmitter and the receiver were on the same side, at a distance of 1.6 m and a height of 1.5 m. The experimenter conducted gesture variations on the opposite side of the receiver and the transmitter, at a distance of 8 m from the transceiver. The experimental setup is shown in Figure 6.

Data collection: In this experiment, six students participated in data collection. Each student performed six gestures indoors 40 times each, resulting in a total of 1440 gesture samples. The gesture types included: push, pull, slide up, slide left, slide right, and wave left and right.

## 6. Result

### 6.1. The Effectiveness of Data Processing Methods

The comparative experimental results highlight the crucial role of the proposed data processing method in improving gesture recognition accuracy. When applied to the processed dataset, all tested gestures showed significant accuracy improvements, especially for dynamic gestures (e.g., “push” and “pull”) and complex gestures (e.g., “slide left” and “slide right”). These results demonstrate the effectiveness of the proposed method in mitigating static vector interference and noise within the original LoRa data.

Static vectors in the raw data presented a major challenge to recognition accuracy, particularly for gestures with similar characteristics, such as “slide up” and “slide left”, leading to frequent misclassifications. By using the proposed data processing technique, static vector interference was effectively removed, allowing the model to focus more accurately on gesture-specific features, thus improving classification performance. Additionally, for dynamic gestures like “wave left and right”, noise in the raw data significantly degraded performance. The proposed method enhanced the dynamic features of these gestures, leading to substantial accuracy improvements.

Furthermore, the consistent improvements across different network architectures—AlexNet, VGG16, and ResNet—validate the robustness and generalizability of the proposed method. This consistency suggests that the data processing technique is versatile and effective across various models.

Despite these promising results, some variations in recognition accuracy among different gestures were observed. These discrepancies may be due to the inherent complexity of certain gestures, differences in signal characteristics, or variations in data collection conditions. Future work will focus on refining the data processing pipeline to improve recognition accuracy, particularly for complex gestures, and exploring multimodal data fusion to further enhance performance and adaptability. The experimental results are shown in Figure 7.

### 6.2. The Generalization of the System

The experimental results demonstrate the system’s robustness and generalization capability when tested on new datasets and under different environmental conditions. Initially, data from a seventh participant were introduced and validated across various scenarios, including dynamic and noisy environments. The system maintained high recognition accuracy, highlighting the effectiveness of its data preprocessing and adaptive segmentation, as well as the feature extraction capability of the SS-ResNet50 model in handling diverse data distributions and complex scenarios.

Building on this, three new gesture types—clockwise rotation, counterclockwise rotation, and push–pull sequence—were added to evaluate the system’s adaptability to more complex hand movements. These gestures, along with the original six, were used to retrain and test the model. The results showed that the system achieved an average recognition accuracy of 94.2% for the new gestures, demonstrating its robustness and adaptability to increased complexity. The experimental results are shown in Figure 8.

The tests also covered different environmental conditions, such as varying levels of background noise and signal reflection. The system showed minimal performance fluctuations, further validating the anti-interference properties of LoRa signals and the robustness of the data processing methods. Even in dynamic and noisy environments, the system consistently delivered high accuracy, showcasing its potential for real-world applications. For the new gestures, while minor misclassifications occurred—mainly between gestures with overlapping features, such as clockwise rotation and counterclockwise rotation—the overall classification performance was still highly satisfactory. The experimental results are shown in Figure 9.

These findings indicate that the system is capable of accommodating more users and handling more complex gestures, emphasizing its scalability and practical applicability. However, there is still room for improvement in enhancing robustness and accuracy for larger user groups and more intricate gestures. Future work will focus on exploring the system’s real-time performance and scalability in multi-user scenarios to support broader application requirements.

### 6.3. More Conforming to the Academic Scenario

After validating the effectiveness of our data processing method, we further evaluated the performance improvements brought by our proposed SS-ResNet model. In this analysis, the ResNet-50 model was used as a baseline due to its solid performance in the previous experiments. Both models were trained on the same dataset, and their recognition rates were compared.

The results show that the SS-ResNet model significantly outperforms ResNet-50 in terms of gesture recognition accuracy. The improvement demonstrates that the integration of dynamic convolution in the SS-ResNet model enhances its ability to extract and classify gesture-related features more effectively. This suggests that the proposed model is better suited for handling the variations and complexities present in LoRa-based gesture data.

In addition to recognition accuracy, the generalization ability of the two models was tested using unseen validation data collected under different environmental conditions. The SS-ResNet model consistently achieved higher recognition rates, confirming its adaptability to new datasets and its robustness in diverse environments. These findings highlight the potential of the SS-ResNet model for real-world applications where robustness and adaptability are critical. The experimental results are shown in Figure 10.

Overall, the results validate the improvements made to the baseline architecture and underscore the importance of designing models tailored to the unique characteristics of LoRa signal data. Future research can build on these findings by exploring further optimization of the model for real-time applications and scaling the system to recognize a broader range of gestures.

### 6.4. Impact of Detection Range

To evaluate the system’s monitoring range, we conducted tests at distances of 8.5 m, 9 m, 9.5 m, 10 m, and 10.5 m. Six gestures were performed at each distance, and their recognition was analyzed using the enhanced SS-ResNet model. The results show a gradual decline in recognition accuracy as the distance increased, which aligns with expectations given the attenuation of LoRa signals over longer ranges and the increased influence of environmental noise. However, even at the maximum tested distance of 10.5 m, the system maintained a relatively high average recognition rate, reflecting its strong robustness and adaptability.

The gradual decline in accuracy can be attributed to several factors. As the distance increases, signal strength diminishes, leading to a reduction in the signal-to-noise ratio. This makes it more challenging for the system to differentiate subtle gesture-induced variations from background noise. Additionally, multipath effects caused by reflections and interference become more pronounced at longer ranges, further complicating the recognition task. Despite these challenges, the system demonstrated its ability to extract meaningful features and maintain stable performance, underscoring the effectiveness of the enhanced model and the proposed data processing techniques. The experimental results are shown in Figure 11.

These results suggest that the system is well-suited for applications requiring long-range gesture recognition, such as monitoring in large indoor spaces or open environments. While the performance is satisfactory across the tested distances, further optimization could focus on improving accuracy at extended ranges. Potential avenues include refining the model’s sensitivity to weaker signals and enhancing noise reduction techniques. These improvements would strengthen the system’s capability to operate reliably in more diverse and challenging environments.

### 6.5. The Impact of Different Modules on the Recognition Effect

In this study, with the aim of clarifying the enhancement effect exerted by each module of the system on gesture recognition performance, a stepwise verification experiment was devised to test the performance contributions of signal processing, activity segmentation, and the improved model (SS-ResNet) in sequence. In the first set of experiments, signal processing was solely conducted on the original LoRa signal, encompassing high-pass filtering, frequency domain analysis, and signal reconstruction, to eliminate static vector interference and low-frequency noise. Subsequently, the ordinary ResNet-50 model was employed for gesture recognition. The outcomes revealed that signal processing significantly mitigated noise interference, with the average recognition accuracy escalating from 78.2% of the original data to 86.5%. Nevertheless, due to the system’s inability to fully extract dynamic features, numerous misclassification phenomena persisted when recognizing dynamic gestures (such as “push” and “pull”) and complex gestures (such as “swipe left” and “swipe right”), indicating that relying solely on signal processing proves insufficient to address the capture of dynamic features. In the second set of experiments, the activity segmentation module was incorporated on the basis of signal processing to undertake dynamic segmentation of the time–frequency domain signal. The activity segmentation module reduced the interference from background noise and static signals by locating the key action regions, significantly enhancing the system’s dynamic feature extraction capability. The experimental results demonstrated that the activity segmentation module raised the average recognition accuracy to 91.4%, particularly in the recognition of dynamic gestures (such as “swing left and right”), where the accuracy rate increased by approximately 6.4 percentage points. Additionally, activity segmentation alleviated the classification confusion resulting from feature overlap. For instance, the misclassification of “swipe up” and “swipe left” was significantly reduced. This outcome validated the crucial role of the activity segmentation module in the analysis of complex gestures. In the third set of experiments, the signal processing and activity segmentation modules were combined, and the improved model (SS-ResNet) was utilized for gesture recognition. SS-ResNet further enhanced the extraction efficiency of gesture features by introducing dynamic convolution and optimized the robustness of feature classification. The experimental results indicated that the average recognition accuracy of the complete system reached 95.8%, exhibiting outstanding performance under all test conditions, especially in long-distance (such as 10.5 m) and complex environments, showcasing stronger adaptability and generalization performance. For example, for complex gestures (such as “rotate clockwise” and “rotate counterclockwise”), the improved model significantly mitigated feature confusion, increasing the accuracy rate by approximately 4.4 percentage points compared to the ordinary model. This suggests that SS-ResNet holds distinct advantages in complex signal analysis and environmental adaptability. To sum up, the signal processing module laid the foundation for subsequent processing by providing cleaner input signals. The activity segmentation module significantly enhanced the analytical capability of complex gestures by focusing on dynamic features, and the SS-ResNet improved model further optimized the overall performance of the system. The organic combination of the three effectively enhanced the stability and accuracy of the system under diverse test conditions. In particular, the introduction of the activity segmentation module and the SS-ResNet model constitutes the core innovation that distinguishes this system from existing approaches, fully verifying their significant value in complex gesture recognition. The experimental results are shown in Figure 12.

## 7. Conclusions

Although the validation of the system was accomplished in a laboratory setting in this study, its technical attributes endow it with the potential for application in more complex scenarios, such as remote gesture control in dynamic environments and even emergency situations like gesture-based distress calls in underground mines. In future studies, we plan to further optimize the real-time performance and adaptability of the system to complex scenarios, and concurrently explore the application of multi-user environments and more fine-grained gestures (e.g., Morse code) in specific tasks. These endeavors are aimed at further enhancing the practicality and multi-functionality of the system, paving the way for its broader adoption in real-world contexts.

This system is characterized by low power consumption, long-range communication, and adaptability to complex environments, rendering it highly potential in various practical applications. For example, in remote healthcare monitoring, it exploits LoRa to facilitate the remote transmission of health data, offering a real-time and cost-effective solution for monitoring patients in remote areas. It is also applicable for large-scale human activity monitoring in smart buildings or public spaces, being capable of analyzing crowd dynamics and behavior patterns to support smart city initiatives. Furthermore, the system is well-suited for low-power environmental sensing in resource-constrained scenarios, such as underground mines, where real-time monitoring of workers’ activities can enhance safety and mitigate accident risks.

Our experiments have further substantiated the system’s adaptability and generalization ability across different scenarios and devices, establishing it as a universal solution for real-world applications. Additionally, this study showcases the potential of LoRa technology in long-range and low-power human activity recognition, particularly in environments demanding strong anti-interference capabilities and adaptability. Long-range recognition is especially valuable in circumstances where traditional communication methods are impeded by physical barriers or significant distances between nodes.

To address these limitations, future work will focus on the following directions:Expanding the system’s capability to recognize more complex gestures and validating its scalability in multi-user scenarios.Enhancing real-time performance by optimizing processing latency and computational efficiency.Exploring multimodal data fusion, such as integrating video or inertial sensor data, to further improve recognition accuracy and system robustness.Validating the system’s performance in unstructured and outdoor environments to ensure its generalizability for diverse application scenarios.

Despite this progress, the system still has certain limitations. The current implementation is wholly reliant on LoRa signals, which might restrict its capacity to capture complex or subtle gestures. Moreover, the system’s performance in multi-user environments remains unverified, and real-time optimization is necessary to address latency and computational efficiency issues in time-sensitive applications.

## Figures and Tables

**Figure 1 sensors-25-01446-f001:**
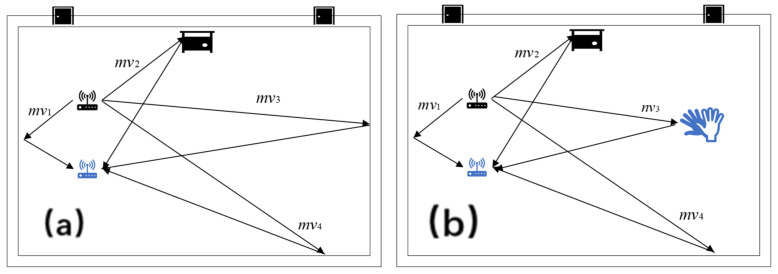
Signal transmission with/without hand gesture changes. (**a**) No hand gesture changes. (**b**) There are hand gesture changes. The black objects are signal transmitters and the blue ones are signal receivers.

**Figure 2 sensors-25-01446-f002:**
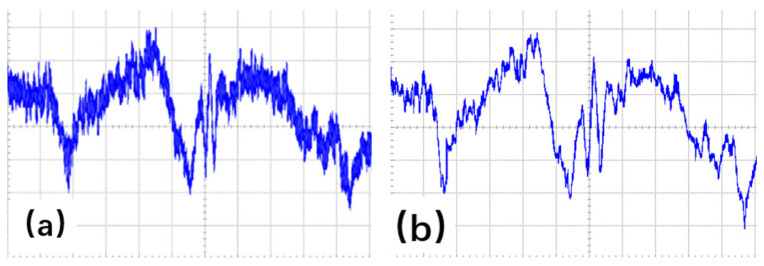
Before and after signal processing. (**a**) Before signal processing. (**b**) After signal processing.

**Figure 3 sensors-25-01446-f003:**
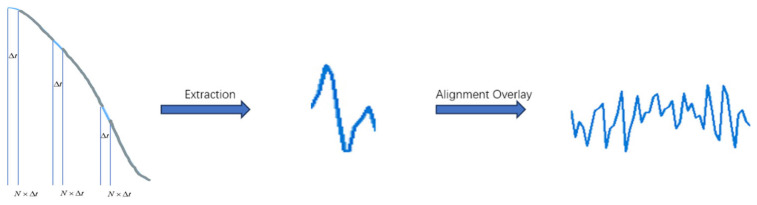
The procedure of data augmentation.

**Figure 4 sensors-25-01446-f004:**
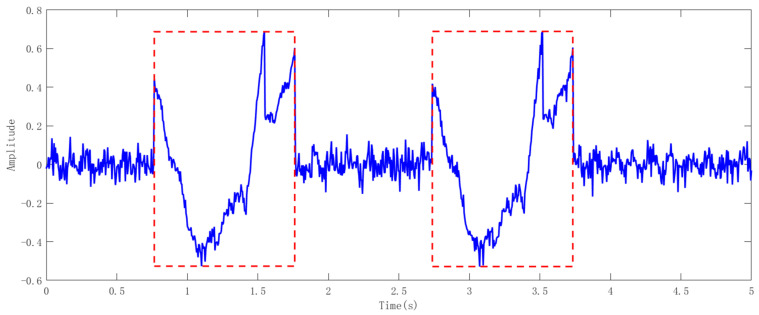
Activity segmentation. The image within the red dotted box is a sliding split image.

**Figure 5 sensors-25-01446-f005:**
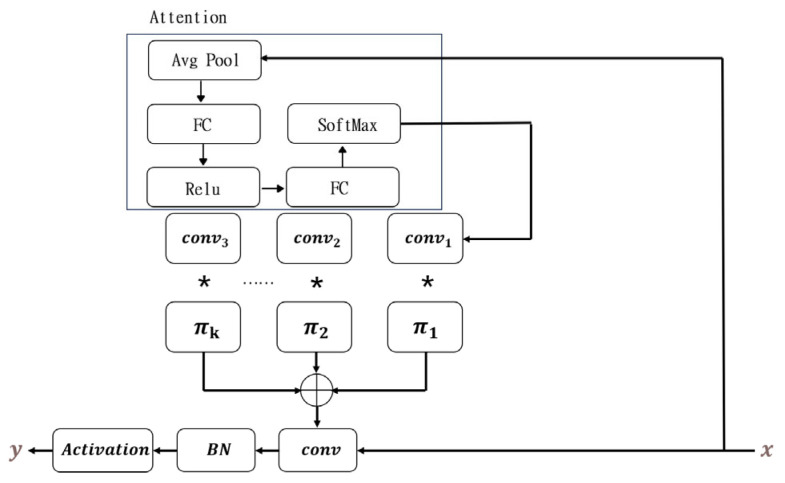
Dynamic convolution module.

**Figure 6 sensors-25-01446-f006:**
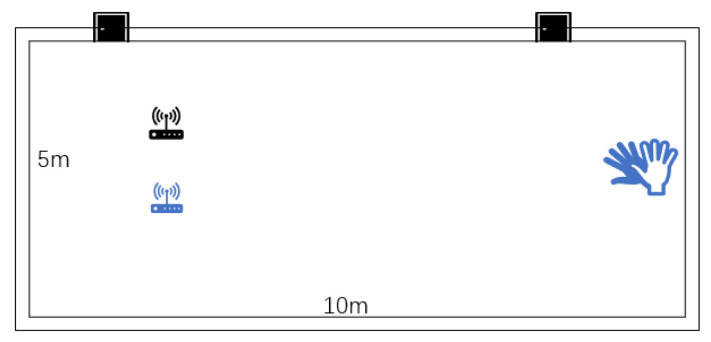
Experimental scenario. The black objects are signal transmitters and the blue ones are signal receivers.

**Figure 7 sensors-25-01446-f007:**
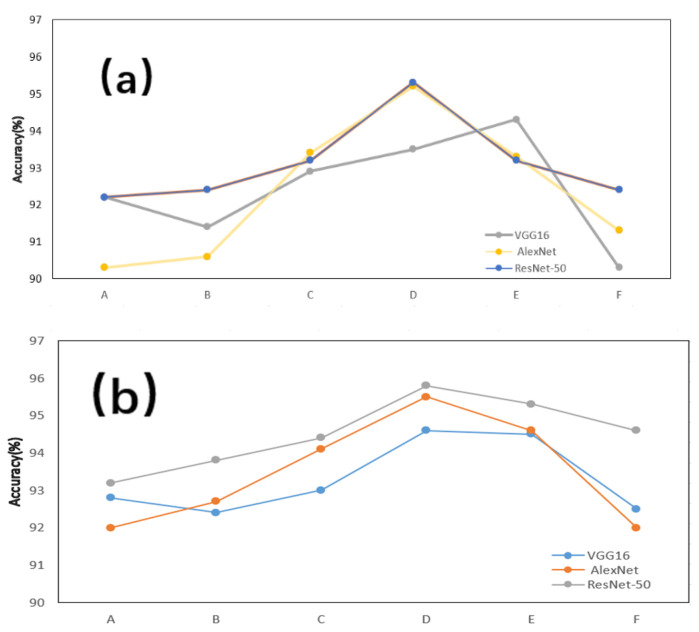
The recognition rates of six gestures for three models. (**a**) Before using LoRaRite. (**b**) After using LoRaRite. A–F respectively correspond to a kind of gesture.

**Figure 8 sensors-25-01446-f008:**
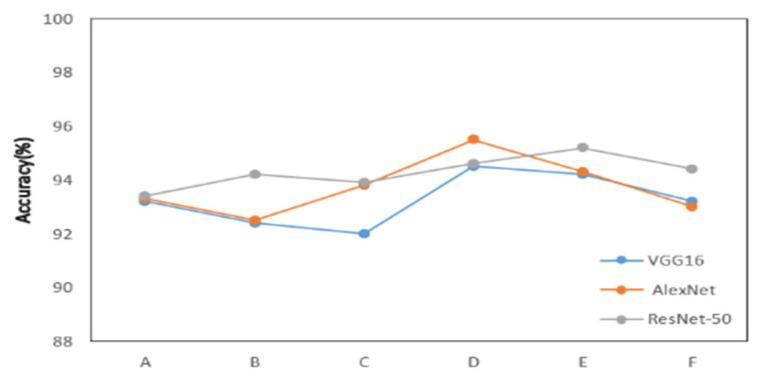
The recognition rates of the new data for the three models. A–F respectively correspond to a kind of gesture.

**Figure 9 sensors-25-01446-f009:**
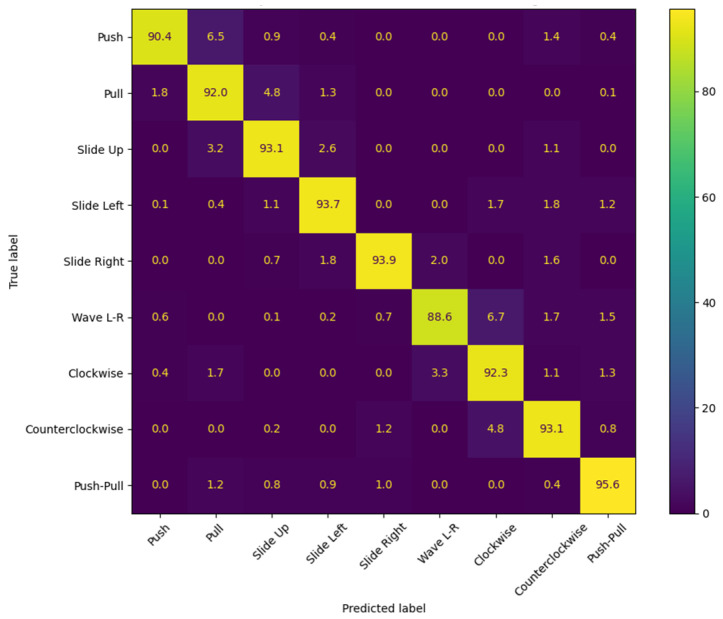
Experimental results after data augmentation.

**Figure 10 sensors-25-01446-f010:**
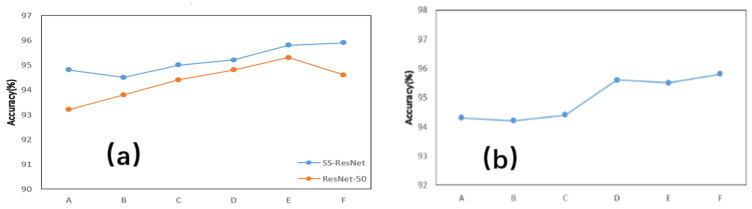
The recognition rates of different models. (**a**) The comparison of recognition. (**b**) The generalization ability of the new model rates of two models. A–F respectively correspond to a kind of gesture.

**Figure 11 sensors-25-01446-f011:**
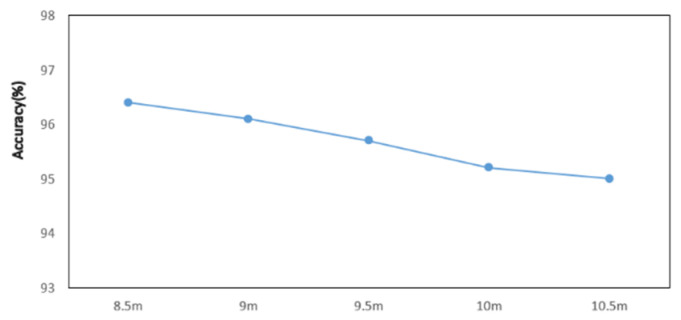
The recognition rates at different distances.

**Figure 12 sensors-25-01446-f012:**
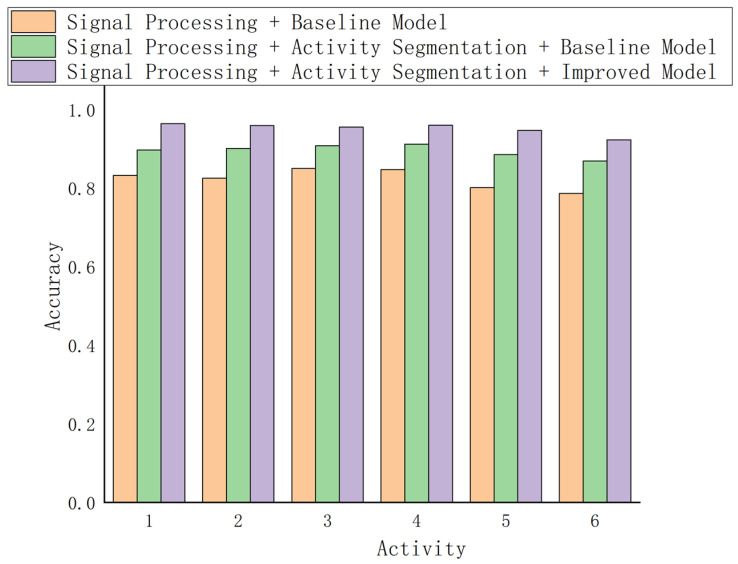
Comparison of the impact of different modules on gesture recognition accuracy.

**Table 1 sensors-25-01446-t001:** Overview of LoRa-based applications.

Study	Application	Approach	Approach
M. Nie [21]	Human activity recognition	STFT + DSP	High accuracy in activity classification and room-level localization
Y. Yin [22]	Distress gesture detection in coal mines	MineSOS system	Reliable communication and gesture detection in complex environments
Mohammad S. Islam [23]	Remote healthcare monitoring	MySignals + LoRa	Automated health data collection, effective in remote healthcare
Yao Ge [24]	Indoor gait recognition	DTW + Phase difference analysis	85.13–84.14% accuracy in LOS, NLOS, and long-distance scenarios
Yuqing Yin [25]	Gait recognition in coal mine environments	Filtering + Autoencoder	96.6–99.7% indoors, 93.3% accuracy at 20 m for eight individuals
Jinkun Han [26]	Posture recognition for smart cities	Multisensor + Random Forest feature selection	95.06–99.38% accuracy, suitable for long-term monitoring

## Data Availability

Data available on request from the authors.

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
