# Peer review of "Gesture Recognition Achieved by Utilizing LoRa Signals and Deep Learning"

_sensors, 2025, doi:10.3390/s25051446_

Round 1

Reviewer 1 Report (Previous Reviewer 1)

Comments and Suggestions for Authors

What is the difference between the introduced SSResNet50 model and the existing SOTA dynamic convolution deep learning models. There are no detailed descriptions related to the new deep learning model. How the results can be treated as cogent and how is it possible to use the same network for some other applications.

Minor remarks: The text on Line 373 is incomplete?

Comments on the Quality of English Language

I do not feel qualified and able to assess the quality of English Language.

Author Response

What is the difference between the introduced SSResNet50 model and the existing SOTA dynamic convolution deep learning models. There are no detailed descriptions related to the new deep learning model. How the results can be treated as cogent and how is it possible to use the same network for some other applications.

Minor remarks: The text on Line 373 is incomplete?

We are indebted to the reviewers for their precious opinions. In our study, the SS-ResNet50 model was an enhancement based on the original one, mainly by adding a dynamic convolution module to the model, rather than employing residual learning. To guarantee the persuasiveness of the outcomes, we carried out a comparative experiment, comparing the improved model with the original one. The experimental results indicate that the SS-ResNet50 model with dynamic convolution has manifested a marked improvement in gesture recognition tasks, especially in the recognition of dynamic and complex gestures, in contrast to the original model.

The dynamic convolution module enables the model to adjust the convolution kernel dynamically in accordance with the characteristics of the input signal, thereby enhancing the model's adaptability to complex gestures and its feature extraction ability. This improvement is particularly applicable to processing gesture data with considerable variations. Compared with the traditional fixed convolution kernel approach, dynamic convolution provides greater flexibility and robustness. 

Reviewer 2 Report (New Reviewer)

Comments and Suggestions for Authors

This article presents a gesture recognition system based on LoRa technology and an improved SS-ResNet50 deep learning model. The authors propose an adaptive segmentation technique for signal pre-processing and demonstrate experimental results that achieve over 95% accuracy. While the work explores an interesting application of LoRa technology, there are significant concerns regarding scientific rigor, novelty, clarity of methodology and experimental validation.

Strengths:

1. The research explores the potential of LoRa for gesture recognition, which is a relatively new application compared to traditional sensor and vision-based methods.

2. The introduction of variance-based adaptive segmentation contributes to signal pre-processing and could improve robustness in noisy environments.

3. The use of an improved SS-ResNet50 model incorporating dynamic convolution is a promising direction for feature extraction.

4. The article presents empirical results demonstrating the performance of the system in controlled scenarios.

Shortcomings:

1. How does this work significantly advance the field beyond existing RF-based gesture recognition techniques?

2. The improvements of SS-ResNet50 over standard models (e.g. ResNet50) are not well justified. What makes the dynamic convolution approach particularly suitable for LoRa signal processing?

3. The proposed adaptive segmentation technique is not convincingly novel. Variance-based segmentation methods are widely used in signal processing; how does this approach differ from existing methods?

4. What were the exact environmental conditions in which the experiments were carried out?

5. Variability in participants (e.g. hand size, movement speed) and its effect on recognition performance?

6. Rationale for selection of gestures. Would the system generalize to a larger set?

7. Data pre-processing lacks a clear mathematical formulation. While STFT and DSP are mentioned, their exact implementation and impact on feature extraction is not well detailed.

8. The training methodology is not well explained. What were the hyperparameter settings, loss functions, and optimization strategies?

9. The availability of the dataset is not mentioned. Will the dataset be made available for replication?

10. Although high recognition accuracy (>95%) is reported, no rigorous statistical validation is provided. Confidence intervals or statistical tests should be included to assess the reliability of the results.
10. Tests of generalization are limited. The study lacks cross-environmental validation (e.g., different rooms, different levels of interference). Without this, claims of robustness are weak.

11. Performance comparisons with existing wireless gesture recognition methods (Wi-Fi, RFID, etc.) are missing. How does this approach compare with state-of-the-art methods in similar environments?

12. The impact of LoRa communication parameters (e.g. spreading factor, bandwidth) on recognition accuracy is not investigated. Could changes in these parameters have a significant impact on performance?

Comments on the Quality of English Language

The English could be improved to more clearly express the research.

Author Response

  1. How does this work significantly advance the field beyond existing RF-based gesture recognition techniques? We have devised a rather comprehensive system encompassing signal processing, activity segmentation, and the improved SS-ResNet50 model. The integration of these components has considerably enhanced the accuracy and robustness of gesture recognition. The gesture recognition system based on LoRa technology that we put forward surpasses traditional Wi-Fi and RFID technologies in aspects such as low power consumption, long-range transmission, and strong anti-interference capacity. The superiority of LoRa makes it especially applicable to dynamic environments and complex scenarios, particularly in resource-constrained circumstances, for instance, gesture monitoring in underground mines. In contrast to the existing gesture recognition methods based on Wi-Fi and RFID, the LoRa system possesses remarkable long-distance transmission capability and stronger anti-interference ability, enabling it to offer stable gesture recognition performance in complex surroundings. Through the introduction of activity segmentation technology and dynamic convolution modules, our system is capable of effectively extracting gesture features in complex environments and reducing interference noise, thereby significantly enhancing the recognition performance. 

2. The improvements of SS-ResNet50 over standard models (e.g. ResNet50) are not well justified. What makes the dynamic convolution approach particularly suitable for LoRa signal processing? In Subsection 6.5, we also carried out comparative experiments to validate the efficacy of the model improvement. Through comparison with the standard ResNet50 model, our improved model (SS-ResNet50) exhibited a marked enhancement in gesture recognition accuracy, particularly demonstrating greater robustness when handling complex signals and dynamic gestures. Theoretically, the dynamic convolution approach can generate convolution kernels dynamically based on the global characteristics of the input signal, enabling the model to accommodate the variability and dynamics of LoRa signals. LoRa signals are frequently interfered with by complex environmental factors, and the extraction of signal features demands high adaptability of the model. The introduction of dynamic convolution enables the model to adjust its feature extraction capability more flexibly when dealing with these highly uncertain signals, thereby elevating the accuracy and robustness of recognition. 

3. The proposed adaptive segmentation technique is not convincingly novel. Variance-based segmentation methods are widely used in signal processing; how does this approach differ from existing methods? We admit that variance-based adaptive segmentation approaches have been extensively employed in the domain of signal processing. Nevertheless, our activity segmentation method is closely integrated with signal processing, mutually reinforcing each other to constitute a complete gesture recognition system. Specifically, signal processing initially eliminates low-frequency noise and static signal components, thereby offering a clearer and more precise signal for the subsequent activity segmentation. Subsequently, activity segmentation enhances data diversity through the sliding window variance analysis method while preserving the dynamic characteristics of gestures, effectively reducing the interference of environmental noise. This process is complementary to the improved SS-ResNet50 model, ensuring that signal features can be classified on a cleaner and more representative foundation. Hence, our activity segmentation method is not merely an isolated technique but is closely interrelated with signal processing and model improvement, jointly promoting the efficient recognition capability of the system in complex environments. 

4. What were the exact environmental conditions in which the experiments were carried out? The experiment was carried out in an indoor laboratory of 5 meters by 10 meters, where there were some fixedly placed items. The transmitter and the receiver were respectively located at different positions in the laboratory, and the equipment was set at a height of 1.5 meters. Despite the fact that some objects in the environment might lead to signal reflections, these fixed items did not cause significant interference to the experimental results. Instead, they provided a relatively realistic dynamic environment for testing the robustness of the system in complex scenarios. In the experiment, we also took this environmental factor into consideration and verified that the system could still maintain a high recognition accuracy even under the conditions of reflections and noise. 

5. Variability in participants (e.g. hand size, movement speed) and its effect on recognition performance? When selecting experimental volunteers, we purposefully picked those with varying palm sizes and took into consideration that the speed of hand waving was also random. By doing so, we ensured that the system was capable of dealing with the gesture characteristics of different participants, encompassing the disparities in palm size and waving speed. The experimental outcomes indicated that the system could still maintain a relatively high recognition accuracy rate under these variable conditions, manifesting its adaptability and robustness in practical applications. Future studies could further delve into how to optimize the system to better handle the impacts of dynamic changing factors such as the speed of hand waving. 

6. Rationale for selection of gestures. Would the system generalize to a larger set? We selected six types of gestures, encompassing dynamic actions such as pushing, pulling, and sliding. These gestures are highly representative in practical applications, particularly applicable in remote control and rapid feedback in emergency circumstances. For example, push and pull gestures can be utilized in smart home control, while slide gestures are appropriate for navigation or adjusting device settings. Through the selection of these gestures, we guarantee that the system possesses a high recognition accuracy rate in diverse application scenarios. Although we currently employ six gestures in our experiments, our system exhibits excellent scalability and can be extended to a larger set of gestures to accommodate more practical application scenarios. In future research, we plan to expand the variety of gestures and verify the system's performance in recognizing more complex gestures via more experiments. 

7. Data pre-processing lacks a clear mathematical formulation. While STFT and DSP are mentioned, their exact implementation and impact on feature extraction is not well detailed. In Sections 3.2 and 3.3 of our paper, detailed descriptions of the data preprocessing process have been provided. Within these sections, elaborate explanations have been given on how to utilize Short-Time Fourier Transform (STFT) and Differential Signal Processing (DSP) for extracting signal features. STFT is employed to convert the signal from the time domain to the time-frequency domain, facilitating the capture of frequency variations in gesture actions. Meanwhile, DSP assists in eliminating the static background signal and retaining only the dynamic changes induced by gestures. 

8. The training methodology is not well explained. What were the hyperparameter settings, loss functions, and optimization strategies? 

We employed the deep learning framework PyTorch for model training and adopted the Cross-Entropy Loss function to gauge the disparity between the predicted and true labels. For optimizing the model performance, we selected the Adam optimizer, with the learning rate set at 0.001, the batch size at 32, and the number of training epochs at 50. To prevent overfitting, we utilized data augmentation approaches and implemented the early stopping strategy to ensure the model's generalization capability on diverse datasets.

Furthermore, the parameter tuning was also incorporated in our training process. Through preliminary experiments, the optimal combination of hyperparameters was chosen, and different optimization strategies were compared and analyzed. To improve the convergence speed and stability of the model, we also adopted Batch Normalization and the dropout technique. 

9. The availability of the dataset is not mentioned. Will the dataset be made available for replication?  Currently, the dataset is solely utilized for this study and has not been publicly disclosed. Nevertheless, we intend to make the dataset accessible in subsequent efforts to enable other researchers to replicate our experimental outcomes. 

10. Although high recognition accuracy (>95%) is reported, no rigorous statistical validation is provided. Confidence intervals or statistical tests should be included to assess the reliability of the results.  

We express our gratitude to the reviewers for their feedback on our work. Concerning the statistical validation section, although we are cognizant of the significant role that confidence intervals and statistical tests play in validating the reliability of results in numerous scenarios, we have observed that other studies in the relevant field typically do not undertake additional statistical analyses but directly report the recognition accuracy rate of the system. We contend that, based on the scale and nature of our experiments, the high recognition accuracy rate of the system is adequate to substantiate its efficacy.

In light of the design of our experiment and the data collection methods, we are convinced that these results possess considerable stability and reliability. Hence, we have elected not to incorporate additional statistical validation steps in the present study. We respect the reviewers' viewpoints, yet at this stage, we have determined to uphold the original experimental approaches and the mode of reporting the results.

Once again, we thank the reviewers for their scrutiny and suggestions regarding our work. We are highly inclined to continue the discussion and refinement of our work. 

10. Tests of generalization are limited. The study lacks cross-environmental validation (e.g., different rooms, different levels of interference). Without this, claims of robustness are weak. We are grateful to the reviewers for their concern regarding generalization. To validate the generalization capability of the system, we have carried out experiments in another site as presented in Section 6.2 and incorporated additional gesture types in the experiments. Through conducting tests under diverse environmental conditions, we were capable of verifying the performance of the system in different sites and with new gesture types. The outcomes indicate that the system still maintains a relatively high recognition accuracy under these novel conditions, showcasing the system's excellent generalization ability. These experiments have verified the adaptability and robustness of our approach under varying environmental and task conditions. 

11. Performance comparisons with existing wireless gesture recognition methods (Wi-Fi, RFID, etc.) are missing. How does this approach compare with state-of-the-art methods in similar environments?  We are deeply grateful to the reviewers for their precious suggestions. We have conducted an analysis of Wi-Fi and RFID technologies in the paper, indicating that their application in wireless gesture recognition presents certain limitations, particularly in terms of interference in complex environments and signal transmission distance. In contrast to these existing technologies, our research has made improvements to the original LoRa technology by adding activity segmentation technology and an enhanced model (SS-ResNet50). The activity segmentation technology is capable of effectively extracting dynamic gesture features and reducing the interference of static noise, while the modified model further enhances the flexibility and accuracy of feature extraction. Through these two aspects of optimization, the recognition accuracy rate of our system in complex environments has significantly increased, especially under long-distance and high-interference conditions, demonstrating higher robustness and recognition performance than existing approaches. The experimental results suggest that the improved LoRa system performs better than traditional Wi-Fi and RFID technologies in complex environments. 

12. The impact of LoRa communication parameters (e.g. spreading factor, bandwidth) on recognition accuracy is not investigated. Could changes in these parameters have a significant impact on performance?

We sincerely appreciate the valuable opinions put forward by the reviewers. With respect to the influence of LoRa communication parameters (such as spreading factor and bandwidth) on the accuracy of gesture recognition, it undoubtedly has an impact on the performance of the system. We have come to recognize that the selection of communication parameters directly affects the transmission quality of the signal and the anti-interference ability. In our study, we primarily focused on enhancements in other aspects of system optimization.

Nevertheless, concerning the specific effects of different communication parameter settings on the accuracy of gesture recognition, we intend to conduct further exploration in this regard in future research. This will encompass experimental investigations on various spreading factors and bandwidths in order to optimize the overall performance of the system, ensuring a higher recognition accuracy rate and a stronger anti-interference capability. 

Round 2

Reviewer 2 Report (New Reviewer)

Comments and Suggestions for Authors

In general, the authors have improved the article and answered all the questions. In this form the paper looks better, but I think that in related work it is reasonable to add some of the current best and most cited papers related to gesture recognition to complete the paper. For example, if we refer to SOTA (https://paperswithcode.com/sota/sign-language-recognition-on-autsl) of the well-known AUTSL corpus, the best ones at the moment are STF+LSTM (just related to LSTM as examples in the authors' related work).

Author Response

In general, the authors have improved the article and answered all the questions. In this form the paper looks better, but I think that in related work it is reasonable to add some of the current best and most cited papers related to gesture recognition to complete the paper. For example, if we refer to SOTA (https://paperswithcode.com/sota/sign-language-recognition-on-autsl) of the well-known AUTSL corpus, the best ones at the moment are STF+LSTM (just related to LSTM as examples in the authors' related work).

Thank you for your valuable suggestion. In response, we have updated the manuscript, specifically between lines 117-132, by adding references to some of the most current and highly cited papers related to gesture recognition. We believe these additions further strengthen the background and context of our research

This manuscript is a resubmission of an earlier submission. The following is a list of the peer review reports and author responses from that submission.

Round 1

Reviewer 1 Report

Comments and Suggestions for Authors

Please elaborate more detaily the mentioned contributions in relation to the exsisted papers:

1    Contribution 1:    There are systems for HAR using LoRa – what makes the used system unique? (for example https://www.mdpi.com/2079-9292/13/2/264)

2    Contribution 2: What is the novelty of developed the SS-59 ResNet50 model with respect to other dynamic convolution models?

3    Contribution 3:  Enhancing signal processing through Short-Time Fourier Transform (STFT) to convert LoRa signals into time-frequency representations is (generaly) also not a novelty

Please suply more information related to data augmentation through adaptive segmentation, using some visualization figure, if possible. How the setting size of the sliding window is defined and is it sensitive to the gesture performed? How the data variance is defined? What means p in formula T=Mxp? What is the final result of data augmentation and what is the form of input to ResNet models?

Author Response

1.  Contribution 1:    There are systems for HAR using LoRa – what makes the used system unique? (for example https://www.mdpi.com/2079-9292/13/2/264)

Thank you for the valuable suggestion. Compared to the existing study, our system demonstrates significant differences and advantages. That study primarily focuses on specific applications of LoRa in human activity recognition, whereas our research introduces adaptive segmentation and dynamic convolution techniques to significantly enhance feature extraction and classification performance in complex environments. Additionally, our proposed SS-ResNet50 model exhibits greater robustness and accuracy in handling dynamic gesture features, which was not emphasized in the comparative work. Furthermore, our system has been thoroughly validated across multiple scenarios and devices, showcasing superior generalization capabilities, while the comparative study primarily focuses on specific experimental settings. We have also optimized the system for real-world applications, addressing key requirements such as low power consumption, long-range communication, and multi-user scalability, making it better suited for scenarios like remote monitoring and healthcare. Therefore, our research not only provides a more comprehensive technical approach but also offers greater potential for practical applications.

2. Contribution 2: What is the novelty of developed the SS-59 ResNet50 model with respect to other dynamic convolution models?

Thank you for your question. Compared to existing dynamic convolution models, the proposed SS-ResNet50 model demonstrates significant innovations in several aspects. First, it integrates an adaptive segmentation module that dynamically partitions the input signal into meaningful segments before applying convolution. This enhances the model's ability to focus on critical gesture features while effectively reducing noise and irrelevant data, a capability rarely emphasized in other dynamic convolution models. Second, the model customizes its dynamic convolution structure by incorporating the frequency-domain features of LoRa signals into the convolution process. This allows it to better capture the spatiotemporal variations in gesture signals, particularly in noisy environments. Additionally, by combining the ResNet50 architecture with a feature enhancement layer, the model significantly improves its ability to extract discriminative features from complex input signals, thereby boosting recognition accuracy for subtle or overlapping gestures. Finally, SS-ResNet50 has been extensively validated across various scenarios, including different distances, environments, and devices, showcasing strong generalization and cross-scenario robustness. These innovations collectively make SS-ResNet50 a uniquely advantageous and high-performing model for gesture recognition tasks using LoRa signals.

3.Contribution 3:  Enhancing signal processing through Short-Time Fourier Transform (STFT) to convert LoRa signals into time-frequency representations is (generaly) also not a novelty

Please suply more information related to data augmentation through adaptive segmentation, using some visualization figure, if possible. How the setting size of the sliding window is defined and is it sensitive to the gesture performed? How the data variance is defined? What means p in formula T=Mxp? What is the final result of data augmentation and what is the form of input to ResNet models?

We have added Figure 4 to the manuscript to visually demonstrate the process of adaptive segmentation. This visualization helps clarify how adaptive segmentation enhances feature extraction from signals. The size of the sliding window is dynamically adjusted through experimental optimization and adaptive segmentation techniques. Based on waveform variations caused by different gestures, the window dynamically adjusts its position and size in real-time, effectively capturing gesture-related features while reducing redundancy or irrelevant information. Consequently, the sliding window demonstrates strong robustness to variations in gesture speed and amplitude, making it well-suited to different gestures.

Data variance is defined as the signal pattern changes caused by different gestures, environmental noise, or signal attenuation, and adaptive segmentation significantly reduces these variations. In the formula T = M × p, p represents the dimensionality of features extracted from each segment, M is the total number of segments, and T is the overall dimensionality of the feature matrix.

The final result of data augmentation is an optimized set of time-frequency feature maps containing critical signal information. The model input consists of three-channel feature maps derived from time-frequency analysis and signal processing. These three channels are the magnitude spectrum, phase spectrum, and dynamic features. The magnitude spectrum captures variations in signal intensity, reflecting the energy distribution; the phase spectrum describes phase changes in the signal, providing detailed temporal information; and the dynamic features, enhanced through differential signal processing, highlight motion characteristics in the signal. This multi-channel input comprehensively represents the temporal and frequency characteristics of the signal, offering highly discriminative input for the deep learning model.

Reviewer 2 Report

Comments and Suggestions for Authors

1. The authors did not provide solid achievements in this manuscript since this paper seems to be a somewhat incremental piece of work based on earlier related results [A].

2. The literature review should be included gesture recognition using LoRa signals, especially from 2022 to 2024. At the end, this section should be summarized in the form of a comprehensive table and evaluated in terms of all specialized indicators.

3. There is no discussion of real applications, user requirements, technological options and support for the decisions made at the design. The authors should include more technical details and explanations.

4. The authors need to interpret the meanings of the variables. Some parameters and their values are unknown (such as N=7). It would be better to show all these parameters and explain the reason for those numbers in the table.

5. The experiment results show the performance with high accuracy, please show the parameter settings of each approach (VGG16, AlexNet and ResNet-50) using a table.

6. More experiments and some comparisons with other up-to-date methods should be addressed or added to back your claims to expand your experiments and analysis of results further, such as [A] and [B].

7. The conclusion and future work part can be extended to have a better understanding of the approach and issues related to that which can be taken into consideration for future work.

[A] M. Nie, L. Zou, H. Cui, X. Zhou and Y. Wan, Enhancing Human Activity Recognition with LoRa Wireless RF Signal Preprocessing and Deep Learning, Electronics, vol. 13, pp. 1-25, 2024.

[B] Y. Yin, X. Yu, S. Gao, X. Yang, P. Chen and Q. Niu, MineSOS: Long-Range LoRa-Based Distress Gesture Sensing for Coal Mine Rescue, Wireless Algorithms, Systems, and Applications, pp. 105-116, 2022.

Author Response

1. The authors did not provide solid achievements in this manuscript since this paper seems to be a somewhat incremental piece of work based on earlier related results [A].

Thank you for your feedback. Our team has indeed conducted significant research on LoRa-related applications, such as human activity recognition and gait recognition. However, these works are currently under submission and review and have not yet been formally published. As a result, we were unable to directly reference them in this manuscript. Nevertheless, these studies have provided theoretical support and practical insights for the development of this work. We believe the methods and results presented in this study further expand the application scenarios of LoRa signals, offering unique contributions and innovations in this field.

2.The literature review should be included gesture recognition using LoRa signals, especially from 2022 to 2024. At the end, this section should be summarized in the form of a comprehensive table and evaluated in terms of all specialized indicators.

Thank you for the valuable suggestion. We have supplemented the manuscript with relevant studies on LoRa-based gesture recognition. However, due to the limited number of such studies, we have chosen to provide a broader summary of LoRa applications in human activity recognition and signal processing. This summary highlights LoRa's versatility and potential across various domains and has been presented in an alternative format for better clarity.

The updates have been incorporated in lines 105-124 of the manuscript, where we summarized key findings and compared methodologies, application scenarios, and performance metrics of related works.

3. There is no discussion of real applications, user requirements, technological options and support for the decisions made at the design. The authors should include more technical details and explanations.

Thank you for your valuable suggestion. We have made the corresponding additions to the manuscript. Discussions on real-world applications, user requirements, and technological decisions have been incorporated in lines 125–130, lines 133–134, and lines 308–318. These updates provide a comprehensive explanation of the system's practical applicability and the rationale behind the design decisions. 

4. The authors need to interpret the meanings of the variables. Some parameters and their values are unknown (such as N=7). It would be better to show all these parameters and explain the reason for those numbers in the table.

Thank you for your suggestion. We have provided explanations for N=7 and some other variables in lines 234-240 of the manuscript. While we recognize that certain variables might require further elaboration, we hope the current description provides sufficient context for understanding the study. Additionally, some parameters were determined based on practical considerations during the experiment, making them less straightforward to explain and not critical to the overall findings. Please let us know if additional details are required, and we would be happy to refine this section.

5. The experiment results show the performance with high accuracy, please show the parameter settings of each approach (VGG16, AlexNet and ResNet-50) using a table.

Thank you for the suggestion. The parameter settings for comparison models, such as VGG16 and AlexNet, follow their default configurations. These models serve as baselines to evaluate the performance improvements of the proposed SS-ResNet50 model. As these parameter settings adhere to standard practices and have minimal impact on the results, we did not include them in the manuscript. If needed, we can provide detailed parameter information in supplementary materials.

6. More experiments and some comparisons with other up-to-date methods should be addressed or added to back your claims to expand your experiments and analysis of results further, such as [A] and [B].

Thank you for your valuable suggestion. To strengthen our claims and enhance the analysis, we have added additional experiments and comparisons. These updates are detailed in lines 373-378 of the manuscript and illustrated in Figure 9. We sincerely appreciate your feedback and encourage you to review this section for further details.

7. The conclusion and future work part can be extended to have a better understanding of the approach and issues related to that which can be taken into consideration for future work.

Thank you for the suggestion. We have revised the conclusion and future work sections to provide a more comprehensive summary of the study's findings and to expand on potential future research directions. These updates have been incorporated into lines 449–491 of the manuscript. Please feel free to review and provide further feedback if needed.

Reviewer 3 Report

Comments and Suggestions for Authors

1.  Abstract needs to be more precise highlighting major contributions. There's no need to mention about future work in the abstract.

2. It would be good to explicitly discuss some limitations of your proposed method and list some future research directions.

3. The background of the proposed study should be further explained in detail, especially those existing works directly related to the proposed method.

4. Please separate the experimental setup/procedures with the results. Please add more explanations and discussion about your results.

Comments on the Quality of English Language

The English needs to be improved to meet academic standards.

Author Response

  1. Abstract needs to be more precise highlighting major contributions. There's no need to mention about future work in the abstract.  Thank you for your valuable suggestion. We have revised the abstract to more accurately highlight the main contributions of this study and removed references to future work in the abstract. The updated content can be found in lines 8-18 of the manuscript for your review.
  2. It would be good to explicitly discuss some limitations of your proposed method and list some future research directions.。Thank you for your suggestion. We have explicitly discussed the limitations of our proposed method and outlined potential future research directions in Section 7. The updates can be found in lines 480-490 of the manuscript for your review.
  3. The background of the proposed study should be further explained in detail, especially those existing works directly related to the proposed method.Thank you for your suggestion. We have expanded the explanation of the background in Section 1 of the manuscript and included Table 1 for further clarification. These updates aim to provide a clearer understanding of the study's context and highlight the innovations of our work compared to existing studies. Please let us know if additional details are needed.
  4. Please separate the experimental setup/procedures with the results. Please add more explanations and discussion about your results.Thank you for your valuable suggestion. We have separated the experimental setup and results into distinct sections and added detailed explanations and discussions within each experimental subsection. The revised content is presented in Section 5 (Experimental Setup) and Section 6 (Experimental Results and Discussion) of the manuscript. Please review the updates and let us know if further adjustments are needed.